

# Evaluation of a lower-powered analyser and sampling system for eddy-covariance measurements of nitrous oxide fluxes

Shannon E. Brown[1], Steve Sargent[2], Claudia Wagner-Riddle[1]

[1]School of Environmental Sciences, University of Guelph, Guelph, Ontario, Canada
[2]Campbell Scientific Inc., Logan, Utah, USA

*Correspondence to*: Shannon E. Brown (sbrown06@uoguelph.ca)

**Abstract.** Nitrous oxide ($N_2O$) fluxes measured using the eddy-covariance method capture the spatial and temporal heterogeneity of $N_2O$ emissions. Most closed-path trace-gas analyzers for eddy-covariance measurements have large-volume, multi-pass absorption cells that necessitate high flow rates for ample frequency response, thus requiring high-power sample
pumps. Other sampling system components, including rain caps, filters, dryers, and tubing can also degrade system frequency response. This field trial tested the performance of a closed-path eddy-covariance system for $N_2O$ flux measurements with improvements to use less power while maintaining the frequency response. The new system consists of a thermoelectrically cooled tunable diode laser absorption spectrometer configured to measure both $N_2O$ and carbon dioxide ($CO_2$). The system features a relatively small, single-pass sample cell (200 ml) that provides good frequency response with a lower-powered pump
(~250 W). A new filterless intake removes particulates from the sample air stream with no additional mixing volume that could degrade frequency response. A single-tube dryer removes water vapor from the sample to avoid the need for density or spectroscopic corrections, while maintaining frequency response. This eddy-covariance system was collocated with a previous tunable diode laser absorption spectrometer model to compare $N_2O$ and $CO_2$ flux measurements for two full growing seasons (May 2015 to October 2016) in a fertilized cornfield in Southern Ontario, Canada. Both spectrometers were placed outdoors
at the base of the sampling tower demonstrating ruggedness for a range of environmental conditions (minimum to maximum daily temperature range: -26.1 to 31.6°C). The new system rarely required maintenance. An in situ frequency response test demonstrated that the cutoff frequency of the new system was better than the old system (3.5 Hz compared to 2.30 Hz), and similar to that of a closed-path $CO_2$ eddy-covariance system (4.05 Hz) using shorter tubing and no dryer that was also collocated at the site. Values of the $N_2O$ fluxes were similar between the two spectrometer systems (slope = 1.01, $r^2$ = 0.96);
$CO_2$ fluxes as measured by the short-tubed eddy-covariance system and the two spectrometer systems correlated well (slope = 1.03, $r^2$ = 0.998). The new lower-powered tunable diode laser absorption spectrometer configuration with the filterless intake and single-tube dryer showed promise for deployment in remote areas.





## 1 Introduction

The concentration of $N_2O$ in the atmosphere is rising and is of concern as $N_2O$ has 298 times the global warming potential of $CO_2$ (IPCC, 2013). Agricultural systems contribute a significant proportion of total global $N_2O$ emissions due to microbial soil processes (Davidson, 2009). The main drivers of microbial $N_2O$ emissions are soil conditions (e.g., soil oxygen, carbon, and

nitrogen levels), but distal drivers such as nitrogen fertilization, dry-wet, and freeze-thaw cycles also exert control and result in $N_2O$ fluxes ($F_{N2O}$) that are highly variable in time ('hot moments') and in space ('hot spots') (Groffman et al., 2009; Molodovskaya, et al., 2012). This sporadic nature of $F_{N2O}$ necessitates continuous measurements covering areas large enough to capture the spatial heterogeneity of the fluxes to sufficiently quantify total emissions from agricultural systems (Flechard et al., 2007; Jones et al., 2011; Shurpali et al., 2016). The eddy-covariance (EC) technique gives long-term, continuous, and

spatially integrated measurements, which can fully capture $F_{N2O}$. Advances in technology for $N_2O$ analyzers has increased the number of long-term $N_2O$ measurement campaigns using EC methods (Mishurov and Kiely, 2010; Molodovskaya et al., 2012; Merbold et al., 2014; Huang et al., 2014; Rannik et al., 2015; Shurpali et al., 2016; Wang et al., 2016). However, the application has been limited to sites with high-quality power or short-term deployment. Locations with agricultural $N_2O$ emissions are often rural areas where it may be impractical and expensive to install permanent power infrastructure for long-term $F_{N2O}$ EC

flux measurements.

Gas analyzers used in EC systems require fast responses to distinguish the high frequencies of concentration fluctuations (McBean, 1972; Leuning and Judd, 1996). High-frequency concentration fluctuations can be attenuated by line averaging within the sample cell, as well as sample mixing within various system components, such as intake tubing, dryers, filters, and

rain caps (Moore, 1986; Massman, 2000). Analyzer frequency response often represents a significant proportion of the total high-frequency losses (Horst, 1997; Kroon et al., 2010b), but these sampling-system components may also significantly degrade system frequency response (Aubinet et al., 2016). Modern closed-path analyzers for $CO_2$ EC fluxes achieve high frequency response with low-power pumps by using small sample cells (6 or 16 ml) that operate near ambient pressure (Burba et al., 2010; Novick et al., 2013). The volume of sampling-system components is minimized to preserve the frequency response

without excessive flow restriction that would increase power requirements for the pump, or exceed the range of the sample-cell pressure sensor (Aubinet et al., 2016; Ma et al., 2017).

In contrast, all $N_2O$ analyzers tested in a recent EC field intercomparison have large sample cells (~ 500 ml) that must operate at significantly reduced pressure (50 to 120 mb), necessitating high flow rates (12 to 17 L min$^{-1}$) (Rannik et al., 2015). The

high flow rate and low pressure requires the use of a relatively high-power pump (500 to 1000 W). Reducing the volume of the sample cell and sampling-system components is a means of maintaining good frequency response at a lower flow rate while reducing the power requirements of $N_2O$ EC systems. Using a dryer on the intake line may degrade system frequency response, but can eliminate uncertainties associated with spectroscopic corrections for line broadening due to water vapor



(Neftel et al., 2010) and density corrections (Webb et al., 1980). The typically high sample flow rates require the use of multi-tube dryers to achieve an acceptable residual humidity. However, laminar flow within the dryer degrades system frequency response because air velocity varies from the center to the wall of the tube (Leuning and Judd, 1996), and multi-tube dryers can further degrade frequency response if the flow varies among the individual tubes. This problem cannot be overcome by

simply increasing flow because a higher flow would require a larger dryer. Alternatively, analyzers with smaller sample cells require lower sample flow, allowing the use of single-tube dryers and lower-powered pumps to achieve the frequency response required for EC measurements.

An upgraded version of a first-generation tunable diode laser absorption spectrometer (TDLAS) (Edwards et al., 2003) has

recently been made commercially available (TGA200A, Campbell Scientific Inc., Logan, Utah, USA). The previous model was suitable for long-term $N_2O$ flux measurements for a variety of agricultural and forested sites (Scanlon and Kiely, 2003; Pihlatie et al., 2005; Pattey et al., 2006; Mishurov and Kiely, 2010; Molodovskaya et al., 2012). The upgraded model has been designed to reduce power consumption of EC measurement systems by reducing the volume of the sample cell. This was done without changing the path length, thereby preserving measurement precision. The smaller sample cell allows a lower flow rate,

enabling the use of a single-tube dryer that maintains frequency response better than high-capacity, multi-tube dryers. Adapting a new vortex intake—originally developed for use with a $CO_2/H_2O$ EC analyzer (Ma et al., 2017)—to the $N_2O$ analyzer should further improve the frequency response as this device removes particulates from the air stream without adding a significant mixing volume. The vortex intake has the additional advantage of not requiring a traditional filter that can clog over time, restricting flow to the analyzer, and requiring maintenance. This is an important aspect of EC system design for long-term

studies because, as stated by Nelson et al., 2004, all components need to be sufficiently rugged to withstand exposure to environmental conditions and run reliably for long time periods with minimal maintenance.

Cutoff frequencies are often used to characterize the frequency response of EC analyzers and systems (McBean, 1972; Nelson et al., 2004; Detto et al., 2011; Rannik et al., 2015) and for correcting flux values for high-frequency losses (Moore, 1986;

Horst, 1997; Aubinet et al., 2000; Massman, 2000; Massman and Lee, 2002). Determining frequency response using field-measured cospectra (Aubinet et al., 2000; Ibrom et al., 2007) is challenging for $N_2O$ EC systems as this requires a strong concentration signal occurring simultaneously with environmental conditions appropriate for scaling with temperature spectra. Direct characterizations are more appropriate for $N_2O$ analyzers since emission events may occur for only a few days in a year, reducing the likelihood of ideal conditions for cospectral analyses. Similarly, evaluating system lag times of closed-path $N_2O$

analyzers used for EC flux processing with the standard cross-correlation method requires a strong signal (Kroon et al., 2010b; Neftel et al., 2010). Directly measuring lag times will give a better estimate of the tube delay for periods with low $N_2O$ signals.





We present the first field trial of a new TDLAS with an optimized sampling system, operating at lower flow rates with a lower-powered pump. Continuous $F_{N2O}$ measurements are presented over two growing seasons for an agricultural site in a cold climate. The system was evaluated for overall performance of the instrument in terms of ease of operation, data quality, and suitability for EC measurements. $N_2O$ fluxes measured by the new TDLAS were compared to those measured by the previous

generation TDLAS and $CO_2$ fluxes ($F_{CO2}$) were compared to fluxes measured by a short-tubed $CO_2$ analyzer. A novel method for determining EC system frequency response and lag time was conducted in situ to determine the analyzer frequency response under field operating conditions.

## 2 Methods

### 2.1 Study Site

Flux measurements took place at the Elora Research Station, Elora, Ontario, Canada (43.3° 27.8' N 80.24° 20.4' W) from May 2015 to October 2016 as a part of a four-plot flux-gradient study on the effect of nitrogen fertilizer management practices on $F_{N2O}$. The four 4-ha plots were located within a larger 30-ha aerodynamically homogeneous and flat field (Fig. 1a) under the same crop management practices. The soil at the site was a Guelph silt loam (fine loamy, mixed, mesic *Glossoboric Hapludalf*). Corn was planted 10 May 2015 and again on 5 May 2016. Urea fertilizer was broadcast-applied at the time of planting in both

years to the plot where instrumentation was set up (Fig. 1a). Corn grew to a maximum height of approximately 2.5 m each summer. Harvest for 2015 started on 27 October and completion was delayed due to rain until 2 November. The field was tilled immediately afterwards, leaving a bare-soil surface from 2 November 2015 to 30 May 2016 when the corn emerged. Harvest in 2016 occurred on 7 October. The field plot was located 400 m from an Environment Canada weather station that measured supporting variables of mean wind direction, air temperature, pressure, and precipitation.

### 2.2 Eddy Covariance Measurements

An EC system (CPEC200, Campbell Scientific Inc., Logan, Utah, USA) was installed in the southwest corner of the 4-ha plot (Fig. 1a, 1c) and was comprised of a sonic anemometer (CSAT3A) to measure three-dimensional wind and sonic temperature, a closed-path infrared gas analyzer (EC155) to measure mixing ratios of $CO_2$ and $H_2O$, and a datalogger (CR3000). The sonic anemometer was oriented towards the predominant wind direction (270°) (Fig. 1b). Two TDLAS were added to this system

(TGA100A and TGA200A, Campbell Scientific Inc., Logan, Utah, USA) (see details below). Each measured mixing ratios of $N_2O$ and $CO_2$. The analyzers include temperature-controlled, weatherproof cases that allowed them to be placed on two cinder blocks set on the ground at the base of the tower (Fig. 1c). Details on the operation of each gas analyzer are given below. The EC measurements were made at a height of 2.0 m when the surface was bare; the intakes and sonic anemometer were raised to 1.75 m above the canopy top throughout the growing seasons. Data from the sonic anemometer and all three analyzers were

recorded at a sampling rate of 10 Hz. Each gas analyzer included a user-configurable digital filter set for a 5 Hz cutoff frequency to avoid aliasing.



### 2.3 Gas Analyzers

The operating principles of the TDLAS used can be found in Edwards et al. (2003) and Wagner-Riddle et al. (2005). Briefly, a temperature-controlled laser emits a beam at an absorption wavelength of the gas species of interest. This beam is split and directed through the sample cell as well as a reference cell supplied with a known reference gas. The laser wavelength is swept

across an individual absorption line by adjusting the laser current. This produces the absorption spectrum, measured by detectors at the ends of the reference and sample cells. The mixing ratio in the sample cell is calculated from the ratio of the spectral absorbance of the reference and sample cells. Both TDLAS are capable of simultaneous measurement of multiple trace gases on separate absorption lines (dual-ramp mode). For periods during this study, the analyzers were set to measure the absorption of $N_2O$ and $^{13}CO_2$.

Table 1 lists the equipment and measurement mode used for each analyzer through the measurement campaign. Operating configurations of $N_2O$ analyzers used in Rannik et al. (2015) are given for comparison. The analyzers included enclosure temperature controllers to maintain temperatures of 20°C in winter and 40°C in summer. Flow from the reference gas cylinders was set at 10 ml min$^{-1}$. Analyzer calibrations were performed shortly after the initial installation and repeated each time a

reference cylinder was replaced. Additional information on calibration and correction of $^{13}CO_2$ to total $CO_2$ can be found in the Supplementary Materials.

The "legacy" TDLAS (TGA100A, manufactured in 2005, hereafter referred to as TDLAS-LN) used a cryogenically cooled, lead salt diode laser operating at 83.5 K and 2243.11 cm$^{-1}$ ($N_2O$) and 2343.585 cm$^{-1}$ ($CO_2$). The TDLAS-LN required twice-

weekly filling of the laser dewar with liquid nitrogen. This analyzer was operated in a dual-ramp mode measuring both $N_2O$ and $CO_2$ throughout the measurement campaign, following a typical air-sampling configuration for EC measurements (Fig. 1d) (see also: Mammarella et al., 2010; Mishurov and Kiely, 2010; Molodovskaya, 2012; Rannik et al., 2015). Air was sampled through a 6 ml rain-shielded intake (part number 17882, Campbell Scientific, Inc., Logan, Utah, USA) and a dryer assembly (PD1000 Campbell Scientific, Inc., Logan, Utah, USA), which included a 2 μm polypropylene filter element in a low-volume,

high-flow filter holder, and a multi-tube dryer (200-tube, 1.2 m Nafion™ dryer element; Perma Pure, Lakewood, New Jersey, USA). A tee fitting and needle valves split the flow between dryer purge (3 L min$^{-1}$) and sample flow directed to the analyzer (14.5 L min$^{-1}$) (Fig. 1d). Tubing between the intake, dryer assembly, and the analyzer was 4.3 mm ID Synflex™ tubing. The total length of tubing (plus dryer) from the intake to the analyzer was 6 m from 10 May 2015 to 8 July 2015, at which time the tubing length was increased to 10 m to accommodate the increase in canopy height. A 950 W rotary vane pump (RB0021,

Busch Vacuum Technics, Inc., Boisbriand, Quebec, Canada) was connected to the analyzer using ~50 m of 25.4 mm ID PVC suction hose (Tigerflex K100, Kuri Tec, Brantford, Ontario, Canada), to draw air through the analyzer and the dryer purge (Fig. 1d). The volume of the sample cell of TDLAS-LN was 480 ml. This setup is similar to the TGA100A used by Rannik et al. (2015), although for the present study the original filter holder (Gelman 1235, Pall Corp.), included as part of the dryer



assembly, was updated to a newer design (part number 20553, Campbell Scientific, Inc., Logan, Utah, USA) with lower volume (5 ml compared to 20 ml). Filter elements were replaced approximately every 2 weeks. The nominal sample cell pressure was 58 mb throughout the measurement campaign.

The new $N_2O$ analyzer (TGA200A, hereafter referred to as TDLAS-TE) was similar to the TDLAS-LN with two major upgrades: a smaller-diameter sample cell and a room-temperature laser. It used a distributed feedback interband cascade laser (DFB ICL, nanoplus Nanosystems and Technologies GmbH, Gerbrunn, Germany) that was cooled thermoelectrically as opposed to with liquid nitrogen. The laser temperature was set to -1°C and the laser current was tuned to measure $N_2O$ at 2237.7 cm$^{-1}$ and $CO_2$ at 2237.3 cm$^{-1}$ from 10 May 2015 to 29 May 2015, when the laser was configured to measure $N_2O$ at
2235.5 cm$^{-1}$ to assess performance of the laser in single-ramp mode for approximately 2 months. On 31 July 2015, the laser was switched back to dual-ramp mode until 31 March 2016. On this date, a new $N_2O$-only laser was installed in the TDLAS-TE. This laser operated at -2.4°C at 2208.6 cm$^{-1}$.

The sample cell of TDLAS-TE has a similar length (1.5 m) compared to the TDLAS-LN, but a smaller inner diameter (12.7
mm compared to 18.8 mm). The smaller inner diameter gives the newer sample cell a volume of 200 ml, a reduction of 280 ml from the original size. From 10 May 2015 to 22 June 2015 the TDLAS-TE sampling system (intake assembly, dryer, and pump) was similar to that of the TDLAS-LN. The TDLAS-TE sampling system was replaced on 24 June 2015 with a new prototype design including a vortex intake and single-tube dryer (Fig. 1e), optimized to take advantage of the smaller sample-cell volume. The intake assembly and filter were replaced with a vortex intake that removes particulates without a traditional
filter (U.S. Patent No. 9,217,692). The vortex intake was adapted for this application by extending the length of the sample tube (2.2 mm ID stainless steel) to 1.1 m, and by providing a separate high-capacity filter (Numatics PXB-02, ASCO Numatics, Florham Park, New Jersey, USA) and diaphragm pump (E163-11-120, Parker-Hannifin, Inc. Hollis, NH) for the bypass flow. On 24 December 2015 this pump was replaced by one designed for long-term continuous use (L061B-11, Parker-Hannifin, Inc. Hollis, New Hampshire, USA). The multi-tube dryer was replaced with a prototype design using a single Nafion® tube
(2.2 mm ID, 7.3 m long TT-110, Perma Pure LLC, Lakewood, New Jersey, USA). The tube was housed in a large-diameter (25.4 mm) flexible shell to allow it to be purged in reflux mode with little pressure drop. A needle valve at the outlet of the dryer controlled the sample-flow rate to 3.5 L min$^{-1}$. The total length of the intake assembly and dryer was 9.1 m. A short (0.1 m) tube (4.3 mm ID Synflex$^{TM}$) connected the needle valve to the analyzer inlet. A lower-powered (250 W) dry scroll pump (nXDS6i, Edwards Ltd., West Sussex, United Kingdom) pulled the sample air through the intake assembly, dryer, and analyzer
via approximately 50 m of 25.4 mm ID PVC suction hose (Tigerflex K100, Kuri Tec, Brantford, Ontario, Canada) (Fig. 1e). The nominal sample cell pressure was 59 mb at the start of the measurement campaign and set to 35 mb after the new sampling system was installed on 24 June 2015.





CO$_2$ and H$_2$O fluxes were measured with a closed-path EC system (CPEC200) that included the gas analyzer (EC155), sonic anemometer (CSAT3A), as well as the sample pump, enclosures, and mounting hardware. Details on the EC155 analyzer can be found in Novick et al. (2013). This CPEC200 system was configured with the manufacturer's valve module for automatic zero and span, as well as a scrub module to supply dry, CO$_2$-free air for setting zero on the analyzer (see details in

Supplementary Materials). The system was configured to automatically set the analyzer to zero and CO$_2$ span daily at 1:00 AM. The EC155 was reconfigured on 10 February 2016 with the production version of the vortex intake/sample cell prototype that was tested by Ma et al. (2017). Nominal flow settings of 8 L min$^{-1}$ total flow, with 6 L min$^{-1}$ through the EC155 sample cell and 2 L min$^{-1}$ for the vortex bypass, were used.

### 2.4 Frequency Response Test

The impulse response method (Sargent, 2012) was used in situ to measure the frequency response of each EC system, and derive the tube delay ($t_{del}$), cutoff frequency ($f_c$), and effective time constant ($\tau_e$). Briefly, this method entails injecting an impulse of high N$_2$O and CO$_2$ concentration into the sample air stream. Measuring the response to this impulse and taking the Fourier transform of this result gives the frequency response. This gives a conservative estimate of the frequency response of the system, as any attenuation of high frequencies caused by the test method are included in the system frequency response.

The impulse responses were measured in situ for each analyzer by injecting high-concentration (2500 ppm N$_2$O and 30000 ppm CO$_2$) gas directly into the intake with a fast-acting solenoid valve (VHS micro dispense valve, The Lee Co., Westbrook, Connecticut, USA) driven with a spike-and-hold driver (part number IECX0501350AA, The Lee Co.) every 10 s, as controlled by a datalogger. For the TDLAS-TE and EC155, which were equipped with the vortex intake, the dispense valve nozzle was

inserted into the rain cap end for a test of the complete system. The TDLAS-LN used a rain cap that did not permit this direct injection. This rain cap was removed and replaced with a small adapter to mount the valve. The dispense valve was driven open for less than 5 ms for each pulse. For the N$_2$O analyzers, frequency response was measured at various pressures, including the standard operating pressure (TDLAS-LN: 29, 38, 45, and 58 mb; TDLAS-TE: 28, 30, 35, and 43 mb). These pressures were achieved by adjusting the needle valve that sets the sample flow. The EC155 frequency response was measured with the

production version of the vortex-intake assembly. The concentration data and the voltages to the solenoid valve were recorded at 20 Hz for 20 min per test.

The impulse response was calculated by subtracting the average ambient background N$_2$O or CO$_2$ concentrations from the data, overlaying the multiple 10 s periods, and then taking an average of the responses. The $t_{del}$ was calculated as the delay

from the valve-open signal to the peak of the impulse response, minus processing time for the analyzer (750, 372, and 800 ms, respectively, for TDLAS-LN, TDLAS-TE, and EC155). This tube delay represents the physical travel time through the sampling system and into the analyzer sample cell.




The frequency response was calculated as the normalized amplitude of the Fourier transform of the impulse response. The $f_c$ was determined graphically as the frequency at which the frequency response reached a value of $1/\sqrt{2}$. Following common practice, we compared this frequency response to a linear, first-order system (Moore, 1986, Horst, 1997; Massman, 2000, Massman and Lee, 2002, Ibrom, 2007). This transfer function is often related to electronic circuits, but it also describes transfer function of an ideal mixing volume (Horst, 1997). The amplitude of the transfer function ($T_{FR}$) is:

$$T_{FR} = \frac{1}{\sqrt{1 + \left(\frac{f}{f_c}\right)^2}} \tag{1}$$

Alternately, the transfer function may be parametrized by an effective time constant ($\tau_e$), calculated using:

$$\tau_e = \frac{1}{2\pi f_c} \tag{2}$$

Theoretical residence times ($\tau_{res}$) of the analyzer sample cells were calculated based on the sample-cell volume, pressure, and flow rates from Table 1. Theoretical cutoff frequencies ($f_{c,t}$) and corresponding theoretical effective time constants ($\tau_{e,t}$) were calculated from $\tau_{res}$ using different models for multi-pass and single-pass sample cells. Multi-pass sample cells, which tend to be relatively short for a given volume with low length to diameter ratios, are often modelled with the assumption of perfect mixing ($\tau_{e,t} = \tau_{res}$) (Nelson, et al., 2004). In contrast, the EC155, TDLAS-LN, and TDLAS-TE have higher length to diameter ratios and have relatively long, narrow sample cells. For these analyzers the theoretical frequency response is modelled assuming no mixing in the sample cell, which is analogous to line averaging in a sonic anemometer when the sonic path is aligned with the wind vector. The amplitude transfer function for line averaging is (Mitsuta, 1966):

$$T_{LA} = \sqrt{\sin^2(\pi\tau_{res}f)/(\pi\tau_{res}f)^2} \tag{3}$$

This transfer function has an approximate cutoff frequency:

$$f_{c,t} = \frac{2.78}{2\pi\tau_{res}} \tag{4}$$

The corresponding effective time constant from eq. 2 is:

$$\tau_{e,t} = \frac{1}{2\pi f_{c,t}} = \frac{\tau_{res}}{2.78} \tag{5}$$

### 2.5 Flux Calculations

Fluxes were calculated as the covariance between fluctuations of the scalar concentration ($s$) and the vertical wind velocity ($w$):

$$F_s = \overline{w's'} \tag{6}$$

where $s$ is the mixing ratio of either $N_2O$ or $CO_2$.



Fluxes were calculated for 30 min time intervals in Matlab™ using modifications to functions from Sturm et al. (2012). The vertical wind was despiked and then rotated using a double rotation (Aubinet et al., 2012; Kaimal and Finnigan, 1994). The high-frequency concentration data for all analyzers was despiked and filtered according to instrument diagnostic codes. Concentration measurements from both $N_2O$ analyzers were filtered for periods of equipment maintenance or malfunction. All

concentrations measured by TDLAS-TE and the EC155 were linearly detrended. Concentration data from TDLAS-LN were linearly detrended until winter of 2016 when the concentration signal periodically indicated optical fringing of the laser. A Chebyshev filter was used to detrend and remove the artificial low-frequency fluctuations for those 30 min periods. The 30 min flux values were filtered for low wind-speed conditions (friction velocity ($u_*$) < 0.1 m·s$^{-1}$) and wind direction of 60 to 120° (shadow of instrument tower). Vibrations from strong winds against the side of TDLAS-LN caused increased concentration

noise. This occurred during periods when the field was bare and mean wind speeds were greater than 5 m s$^{-1}$ from 200° to 360°. Data from these periods were filtered out. Flux data from TDLAS-TE were not affected by the strong winds as it was sheltered by TDLAS-LN for southwest to northerly wind directions, and data with winds against the side of TDLAS-TE were already filtered out due to tower interference (Fig. 1).

Lag times ($t_l$) for analyzers measuring $CO_2$ concentration (TDLAS-TE when operated under dual-ramp mode; TDLAS-LN and EC155) were calculated using cross-correlation such that $t_l$ was the time delay that maximized the covariance between the concentration and $w$ signals. $CO_2$ concentrations were used for the $t_l$ calculation as the $N_2O$ signal was too weak for cross-correlations for the majority of time periods, as was also observed by Kroon et al. (2010a). This was determined on a day-by-day basis as the conditions that affected concentration signal delays (e.g., pump performance, filter clogging) remained

relatively consistent on a daily time scale. For the daily lag value, $t_l$ was calculated for individual 30 min periods per day with the strongest $F_{CO2}$ as determined from raw 30 min flux values. An average of six lag times for each analyzer was used for a given day. Lag times ranged from 0.5 to 1.0 s for the TDLAS-TE and 0.6 to 1.2 s for the TDLAS-LN; $t_l$ for the EC155 was consistent at 0.1 s. Processing delays in the sonic anemometer and the gas analyzers were subtracted from these lag times to calculate tube delays for comparison with $t_{del}$ derived from the frequency response test. Values for $t_l$ derived from $t_{del}$ and sonic

processing delays were used for periods when the TDLAS-TE was setup to measure only $N_2O$ concentrations.

The $\tau_e$ derived from the frequency response test was used to calculate corrections for high-frequency losses using the simplified formula given by Horst (1997):

$$T_r = \frac{1}{1+\left(2\pi n_m \frac{U}{z}\tau_e\right)^\alpha} \tag{7}$$

where $U$ is the mean wind speed, $z$ is the measurement height, $n_m$ and $\alpha$ are constants determined from the idealized spectra of Kaimal et al. (1972) using values for unstable and stable conditions as presented by Horst (1997).



### 2.6 Flux Comparisons

Half-hourly $F_{N2O}$ and $F_{CO2}$ obtained with the two TDLAS were compared over the whole measurement period, and $F_{CO2}$ from both TDLAS were also compared to the EC155. Agreement was evaluated based on $r^2$ and slope of the linear regression between values from the different analyzers. Cumulative $F_{N2O}$ over the measurement period for each TDLAS were calculated.

No gap filling was applied to the time series, so totals do not represent absolute emissions for this site. Emissions were accumulated using only 30 min $F_{N2O}$ when data were available from both analyzers and converted from ng $N_2O$-N m$^{-2}$ s$^{-1}$ to g $N_2O$-N ha$^{-1}$.

Random errors of each of the individual 30 min $F_{N2O}$ ($\delta_F$) as measured by the TDLAS-LN and TDLAS-LE, were calculated

using the filtering method of Salesky et al. (2012). This method was preferred over those requiring an estimate of the turbulent integral time scale (e.g., Finkelstein and Sims, 2001) as time scales can vary depending on definition (Dias et al., 2004). Salesky et al. (2012) showed that error estimates from the filtering method compared well to those using integral scales. Briefly, this method entailed repeatedly applying filters of increasing width ($\Delta t$) to the high frequency instantaneous fluxes from each 30 min period. Standard deviations of each set of filtered data were fit to $\Delta t^{-1/2}$. These successive power-law fits were then fit

to the entire averaging period to estimate the 30 min $\delta_F$. Propagation of $\delta_F$ was then used to calculate the confidence intervals ($p = 0.05$) of the total cumulative $F_{N2O}$ for the TDLAS-TE and TDLAS-LN time series.

### 3 Results

#### 3.1 N₂O Fluxes and Data Coverage

Measurements of fluxes covered two growing seasons (May to Oct in 2015 and 2016), one post-harvest period (November to December 2015), winter (January to February 2016), and early spring (March to April 2016). Winter was classified as the period when nighttime temperatures were consistently below freezing. Fig. 2 displays the mean daily air temperatures and precipitation for 2015 and 2016. Mean hourly air temperatures ranged from -26.1°C to 31.6°C. Winter of 2016 was on average 3°C warmer than 30-year normals and the maximum snow depth was relatively low at 8 cm. Precipitation totals were close to

30-year normal values for both years; however, May of 2015 received only 17 mm of rainfall until May 31, which was considerably less than the normal amount of 89 mm. Rainfall was lower than normal in May, June and July of 2016 with monthly totals 50% lower than the 30-year monthly normals.

Non-gap-filled mean daily N₂O fluxes obtained with the TDLAS are shown in Fig. 3. One major N₂O flux event occurred in

2015, one day after the first significant rainfall after planting and fertilizer application (8 June 2015 and 9 June 2015, day of year (DOY) 159 and 160 in Fig. 3a). The daily mean 30 min $F_{N2O}$ values for this event as measured by both analyzers were 600 ng $N_2O$-N m$^{-2}$ s$^{-1}$ with the 30 min fluxes during the daytime exceeding 800 ng $N_2O$-N m$^{-2}$ s$^{-1}$ for several hours. Two





subsequent post-fertilizer flux events occurred after rainfall on 16 June 2015 and 28 June 2015 (DOY 167, 179, Fig. 2a) and were smaller in magnitude with daily average fluxes of 100 ng $N_2O$-N $m^{-2}$ $s^{-1}$. Throughout the remainder of the growing season of 2015 and post-harvest period of 2015, mean daily fluxes were small and varied between -10 to 10 ng $N_2O$-N $m^{-2}$ $s^{-1}$. The warmer-than-average winter in 2016 limited flux emissions from freeze-thaw processes that have been typically observed at

this site (Furon et al., 2008; Risk et al., 2013; Abalos et al., 2015; Congreves et al., 2017; Wagner-Riddle et al., 2017) with average daily fluxes ranging from 0 to 30 ng $N_2O$-N $m^{-2}$ $s^{-1}$ (Fig. 3b). The abnormally dry spring in 2016 prevented any significant $N_2O$ flux events associated with fertilizer application. Peak post-fertilizer daily flux values from 5 May 2016 to 13 June 2016 (DOY 126 to 165 in Fig. 3c) averaged 25 ng $N_2O$-N $m^{-2}$ $s^{-1}$. Daily flux values in the summer period of 2016 remained small (~ 5 ng $N_2O$-N $m^{-2}$ $s^{-1}$, Fig. 3c) but predominantly positive throughout the summer.

Both analyzers performed well throughout the measurement period and maintenance requirements were minimal for both. Of the total number of 30 min periods for the whole measurement period, 44% of the 30 min flux values were filtered out due to low friction velocity, wind direction and footprint criteria, warnings from the sonic anemometer from precipitation, disruption from field operations, and power outages. Further filtering based on analyzer operation and diagnostics resulted in an additional

removal of 16% (TDLAS-TE) and 28% (TDLAS-LN) of the fluxes due to equipment failure and periods of equipment reconfiguration. Longer data gaps occurred mostly for the TDLAS-LN due to pump malfunction, a faulty power supply, and an electronic module failure. Data outage for the TDLAS-TE in September 2015 was due to failure of the diaphragm pump providing bypass flow for the vortex intake before a model designed for long-term use was installed. Intermittent periods of data losses for both TDLAS also occurred when the enclosure temperature settings prevented the heaters from maintaining a

constant temperature. This occurred at night during spring and fall periods when differences between daytime and nighttime temperatures were greatest, and midday in the summer months when air temperatures exceeded 30°C.

**3.2 Frequency Response Test**

The in situ frequency response test characterized the frequency response and tube delay for the three analyzers under the

operating pressures tested. Impulse response peaks were at minimum 5 ppm ($N_2O$) and 600 ppm ($CO_2$) above ambient concentrations. The peaks were consistent throughout each of the 20 min tests with an average coefficient of variation of 1.8%. The precision of the impulse responses was further improved by averaging the 120 peaks in each test. Frequency-response curves for each analyzer obtained at the nominal cell pressure (Table 1) are given in Fig. 4. The frequency-response curve for the EC155 drops steeply at 5 Hz as expected due to its digital anti-aliasing filter. The effect of the digital filters is not distinctly

visible for the $N_2O$ analyzers, but all three analyzers have little or no response at or near the Nyquist frequency (10 Hz), giving confidence that the frequency response curves are not contaminated by aliasing. Table 2 contains the corresponding cutoff frequencies and effective time constants. The EC155 showed the best frequency response ($f_c$ = 4.05 Hz and $\tau_e$ = 39 ms) for $CO_2$ (Table 2).  The frequency response to $N_2O$ of TDLAS-TE was nearly as good as the EC155 ($f_c$ = 3.5 Hz, $\tau_e$ = 46 ms),



while TDLAS-LN showed lower frequency response ($f_c$ = 2.3 Hz and $\tau_e$ = 69 ms). TDLAS-LN measured both $N_2O$ and $CO_2$ during the frequency response test; results for $CO_2$ were identical to those for $N_2O$ (data not shown). These measured $f_c$ were used in equation 1 to calculate $T_{FR}$, which are shown for comparison in Fig. 4. All measured $f_c$ values were lower than the best-case theoretical cutoff frequencies ($f_{c,t}$) calculated from $\square_{res}$ (Table 2), indicating some attenuation of high frequencies in the

sampling system components and sample cells, as expected. Frequency responses for both $N_2O$ analyzers were tested at a range of pressures as varying field conditions can cause pressures to drift with pump performance and, in the case of TDLAS-LN, filter clogging. The TDLAS-LN $f_c$ increased significantly with pressure (1.1 Hz at 29 mb to 2.3 Hz at 58 mb), whereas the response of the TDLAS-TE showed less variation (3.1 Hz at 28 mb to 3.65 Hz at 43 mb) (data not shown).

Tube delays determined from the impulse responses at the nominal cell pressure for the three analyzers are given in Table 2. As expected, the EC155 had the shortest $t_{del}$ at 100 ms. The tube delay of TDLAS-TE was longer (578 ms) due to the greater tubing length and lower flow rate. The tube delay for TDLAS-LN was longest at 720 ms, despite its higher flow rate, due to the larger volumes for its tubing, filter, and dryer. Tube delays for the $N_2O$ analyzers over the range of operating pressures showed a decrease in $t_{del}$ with increasing cell pressure (Fig. 5). Decreases in flow rate with pressure for TDLAS-LN acted to

increase $t_{del}$ to a greater degree than for the TDLAS-TE. Tube delays determined from the cross-correlation of $CO_2$ with $w$ ranged from 650 to 1450 ms for TDLAS-LN and 550 to 1050 ms for TDLAS-TE, and were consistent with those determined from the impulse response (Fig. 5).

### 3.4 Flux Comparison

Comparison of $F_{N2O}$ showed similar values for both TDLAS on short and long time scales. Fig. 6a shows the agreement between 30 min $F_{N2O}$ values measured by each analyzer ($r^2$ = 0.97, slope = 1.01). The majority (86%) of 30 min $N_2O$ fluxes were small and below 20 ng $N_2O$-N $m^{-2}$ $s^{-1}$. The general correspondence between the TDLAS-LN and TDLAS-TE flux values remained for the small flux values (Fig. 6b). Scatter in the low $F_{N2O}$ (Fig. 6b) was related to instrument noise. Both analyzers showed excellent agreement with the $CO_2$ fluxes measured using the EC155. The slope and correlation coefficient of the linear

regression between the EC155 and TDLAS-TE were, respectively, 1.03 and 0.998 (Fig. 6c). Similarly, the slope and correlation coefficient of the linear regression between the EC155 and TDLAS-LN were 1.03 and 0.998 (data not shown).

Emissions were accumulated over the measurement period to evaluate the long-term comparison of the total $F_{N2O}$ (Fig. 7). From planting of 2015 to harvest of 2016, TDLAS-TE measured total emissions of 1247 ± 11.27 g $N_2O$-N $ha^{-1}$ and TDLAS-

LN measured 1272 ± 9.27 g $N_2O$-N $ha^{-1}$, where uncertainties are the propagated random errors.



## 4 Discussion

Both $N_2O$ analyzers were well suited for the long-term and continuous measurement of $F_{N2O}$ that are required to fully capture the temporal dynamics of $N_2O$ emissions (Savage et al., 2014; Shurpali et al., 2016). Both operated continuously for over 1.5 years through the various environmental conditions encountered in southern Ontario. Data gaps were caused by problems with

sample pumps, a power supply failure, and from reconfigurations for the intercomparison experiment, (e.g., frequency response testing, swapping lasers, and reconfiguring the sampling systems). Disregarding these time periods, data coverage of 56% was typical for EC measurements at that site after typical EC filtering for rainy periods, low wind speeds, and wind direction (Santos et al., 2011; Brown et al., 2013). Further steps can be taken to reduce data losses attributable to the TDLAS operation, including protecting the analyzer from strong winds, and adjusting the analyzer's internal-temperature setpoint more frequently

(seasonally to monthly).

The older TDLAS-LN system required periodic maintenance including liquid nitrogen fills (twice weekly), replacing filter elements (twice monthly), and changing pump oil (monthly). The TDLAS-TE system with the vortex intake and dry scroll pump required no routine maintenance. The TDLAS-TE also operated using less power for the sample pump than the TDLAS-

LN (250 W versus 950 W). The ruggedness and low maintenance of TDLAS-TE through all seasons showed the suitability of the analyzer for capturing post-fertilizer and freeze-thaw emissions.

The TDLAS-TE system, with smaller sample-cell volume, single-tube dryer, and vortex intake had better frequency response ($f_c$ = 3.5 Hz) than the TDLAS-LN system ($f_c$ = 2.30 Hz), while operating at lower flow rates (3.5 compared to 14.5 L min⁻¹)

and with lower sample pump power consumption. The TDLAS-TE cutoff frequency of 3.5 Hz was only slightly lower than that of the short-tubed EC155 closed-path analyzer ($f_c$ = 4.05 Hz). The frequency response of the TDLAS-TE system was affected less by changes in pressure and flow rate than the TDLAS-LN which helped to maintain constancy of $f_c$ with the slight variations in pressure that occur during long-term field operations. The frequency response of TDLAS-LN was still adequate for EC measurements (McBean, 1972, Eugster et al., 2007). Rannik et al. (2015) reported EC system time constants calculated

from cospectra ($\tau_{e,s}$) for several $N_2O$ analyzers, which are summarized in Table 2. The same model $N_2O$ analyzer (TGA100A) with a similar sampling system at similar flow (17 L min⁻¹) and pressure (50 mb) had a $\tau_{e,s}$ of 120 ms. The lower $\tau_e$ in this study (69 ms) is likely related to the shorter tube and newer lower-volume filter holder as compared to the higher-volume filter holder used in the Rannik et al. (2015) study, or discrepancies between our direct frequency response assessment compared to deriving $\tau_e$ from cospectral analyses. Rannik et al. (2015) found a disagreement of almost a factor of two in the $N_2O$ and $CO_2$

time constants for a system that measured both gases, opting to use the value measured for $CO_2$ for spectral corrections of both gases.





Analyzer sample cell and sampling system design both have an impact on system frequency response. Multi-pass sample cells tend to be much shorter but with larger diameters compared to single-pass sample cells; therefore, the air flow tends to behave as a mixing volume. Measured $\tau_e$ for analyzers with multi-pass sampling cells are generally greater than the $\tau_{res}$ of the sample cell. Nelson, et al. (2004) directly measured the frequency response of an analyzer with a multi-pass sample cell using an

exponential fit to measurements of a step change in concentration. Their $\tau_e$ of 63 ms was slightly greater than the theoretical time constant ($\tau_{res}$ = 54 ms). Neftel et al. (2010) measured the time constant for an EC system with a similar sample cell using the same step-change technique. The measured $\tau_e$ of 400 ms was more than twice the theoretical time constant ($\tau_{res}$ = 175 ms) of the sample cell, likely due to the long tube (30 m) and other sampling components. Single-pass sample cells, which tend to be long and narrow, have less mixing as the air flows through the cell, giving them a theoretical advantage of a factor of 2.78

compared to multi-pass sample cells. The TDLAS-TE $\tau_e$ was within 15% of this theoretical $\tau_{e,t}$, indicating near ideal performance in the sample cell and sampling system. The EC155 $\tau_e$ was a factor of two larger than the theoretical $\tau_{e,t}$. This is likely due to mixing in the sample cell, which has a length/diameter ratio of 15, compared to 110 for the TDLAS-TE. TDLAS-LN has a diameter/length ratio of 75, suggesting its performance should also approach the ideal. However, the TDLAS-LN $\tau_e$ was 40% greater than $\tau_{e,t}$, compared to 15% for TDLAS-TE. This additional frequency response degradation was most likely

caused by attenuation in the multi-tube dryer. Ratios of $\tau_e/\tau_{e,t}$ from similar analyzers operating without driers (Table 2) were comparable to those of the TDLAS-LN and TDLAS-TE systems, although not directly as spectral calculations of $\tau_e$ were used in that study.

Operating the $N_2O$ systems with driers prevented uncertainties from spectroscopic and density corrections. Improper

application of the WPL corrections, as well as variations in flux processing steps, can affect flux values (Aubinet et al. 2012; Mammarella et al., 2016). For fluxes close to zero, WPL corrections for fluctuations in water vapor can be greater than the value of the scalar flux, and uncertainty of the correction can be of similar magnitude to the small fluxes (Detto et al., 2011). Line-broadening caused by moisture in the sample can also cause significant errors in $F_{N2O}$ (Neftel et al., 2010).

Lag times can be difficult to determine using cross-correlation methods on data with weak signals, which is often the case for $N_2O$ (Pihlatie et al., 2005; Neftel et al., 2010). Inaccurate lag times can lead to underestimations of flux values. The $t_{del}$ from the impulse response tests corresponded well with the $t_l$ obtained for $CO_2$ using the cross-correlation method. The $t_{del}$ measured with the impulse response method reduced uncertainty in the $N_2O$ lag times used in the flux processing in the absence of a $CO_2$ signal. This approach is useful for long-term studies measuring agricultural $N_2O$ fluxes when signals are too weak for

automatic covariance maximization and when variations in operating conditions preclude the use of a constant lag value. Tube-delay of the TDLAS-TE (0.6 s at 35 mb) was lower than that of the TDLAS-LN (0.8 s at 58 mb), showing the improved transit time of the upgraded intake system. These values were comparable to tube delay values of analyzers operated without driers



(Rannik et al., 2015), showing how the TDLAS systems used here (with driers) could be optimized for fast response, while also avoiding corrections associated with lack of air drying.

## 5 Conclusions

The field trial of the new TDLAS-TE analyzer with the optimized sampling system consisting of a vortex intake, single-tube drier and lower-power pump demonstrated the ability of this analyzer to operate continuously through several field seasons with minimal maintenance with the frequency response needed for EC measurements. The frequency response was comparable to that of a short-tubed $CO_2$ EC system, and was better than published reports of $N_2O$ EC systems operating without driers. The legacy TDLAS-LN analyzer presented a lower cut-off frequency and higher tube delay than the TDLAS-TE and EC155,

but showed improvement over previous reports due to the use of a shorter intake tube and a lower volume filter holder. The 30 min $F_{CO2}$ and $F_{N2O}$ values measured by the TDLAS systems compared well; $F_{CO2}$ values measured by these systems were also comparable with the EC155.

More studies of long-term, multi-season $N_2O$ fluxes are needed to improve global estimates of agricultural $N_2O$ emissions

(Reay et al., 2012; Wagner-Riddle et al., 2017). The optimal performance combined with its lower power and low maintenance requirements make the TDLAS-TE suitable for $N_2O$ flux measurements under rugged field conditions in remote or rural areas where power quality can be a concern.

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



**Table 1**: Sampling system details for each analyzer throughout the measurement campaign. Changes to equipment during the experimental period are indicated. Operating configurations of $N_2O$ analyzers used in Rannik et al. (2015) are given for comparison.

| | Filter | Dryer | Pump | Flow (L min⁻¹) | Cell Pressure (mb) | Cell Volume (mL) | Tube ID (mm) | Tube Length (m) |
|---|---|---|---|---|---|---|---|---|
| **Current study** | | | | | | | | |
| TDLAS-LN | | | | | | | | |
| before 8 Jul 2015 | 2 μm$^{pp}$ | multi-tube | rotary vane | 14.5 | 58 | 480 | 4.3 | 6 |
| after 8 Jul 2015 | 2 μm$^{pp}$ | multi-tube | rotary vane | 14.5 | 58 | | 4.3 | 10 |
| TDLAS-TE | | | | | | | | |
| before Jun 2015 | 2 μm$^{pp}$ | multi-tube | rotary vane | 14.5 | 59 | 200 | 4.3 | 6 |
| after 24 Jun 2015 | vortex | single tube | scroll | 3.5 | 32 | | 2.2 | 9.1 |
| EC155 | | | | | | | | |
| before Feb 2016 | 20 μm$^{ss}$ | none | diaphragm | 7 | 950 | 6 | 2.7 | 0.6 |
| after 4 Feb 2016 | vortex | | | 6 | 910 | | 2.2 | 0.6 |
| **From Rannik et al. (2015)** | | | | | | | | |
| TGA100A (Campbell Scientific Inc.) | | | | 17 | 50 | 480 | 4 | 17.8 |
| CW-TILDAS-CS (Aerodyne Research Inc.) | | | | 13.2 | 53 | 500 | 4 | 16 |
| N2O/CO-23d (Los Gatos Research Inc.) | | | | 11.6 | 117 | 408* | 8 | 16 |
| QC-TILDAS76-CS (Aerodyne Research Inc.) | | | | 13.5 | 53 | 500 | 4 | 8.5 |

*from LGR user manual
*Notes:*
pp – polypropylene
ss – stainless steel




**Table 2:** Analyzer frequency response characteristics from this study and that of Rannik et al., (2015). Residence times ($\tau_{res}$) were calculated from values given in Table 1. $f_{c,t}$ is the theoretical best-case cutoff frequency, assuming no mixing for the single-pass sample cells and complete mixing for the multi-pass sample cells, and $\tau_{e,t}$ is the corresponding effective time constant. $f_c$ and $\tau_e$ are results from the impulse response test. Results ($\tau_{e,s}$ and $f_{c,s}$) from Rannik et al. (2015) were derived from cospectral analyses. The tube delays $t_{del}$ are the measured total delay minus the processing delay. Time constants and $t_{del}$ are in units of ms and cutoff frequencies are in Hz.

| | $\tau_{res}$ | $f_{c,t}$ | $\tau_{e,t}$ | $f_c$ | $\tau_e$ | $t_{del}$ |
|---|---|---|---|---|---|---|
| **Current study:** | | | | | | |
| TDLAS-LN | 114 | 3.88 | 41 | 2.30 | 69 | 720 |
| TDLAS-TE | 48 | 9.22 | 17 | — | — | — |
| TDLAS-TE (vortex) | 108 | 4.10 | 39 | 3.5 | 48 | 578 |
| EC155 | 54 | 8.19 | 19 | 4.05 | 39 | 100 |
| **From Rannik et al., 2015:** | | | | $f_{c,s}$ | $\tau_{e,s}$ | |
| TGA100A (Campbell Scientific Inc.) | 100* | 4.42 | 36 | 1.33 | 120 | — |
| CW-TILDAS-CS (Aerodyne Research Inc.) | 119 | 1.34 | 119 | 2.27 | 70 | — |
| N2O/CO-23d (Los Gatos Research Inc.) | 244 | 0.65 | 244 | 0.61 | 260 | — |
| QC-TILDAS76-CS (Aerodyne Research Inc.) | 116 | 1.37 | 116 | 1.99/0.94[†] | 80/150[†] | — |

*Assumes 15% of flow is used for purge.

[†]Calculated with $CO_2$

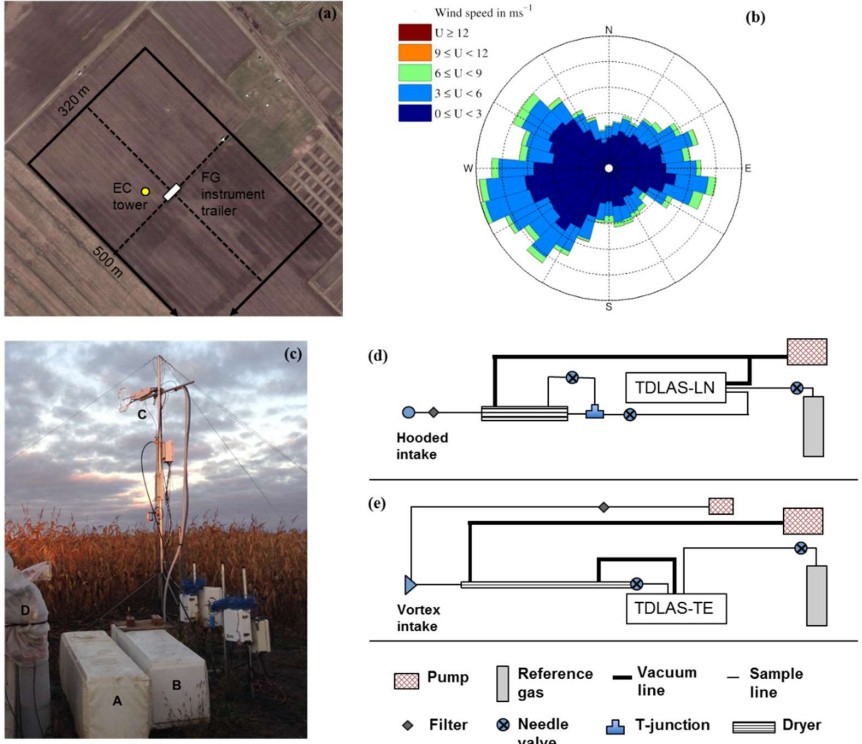

**Figure 1:** Descriptions of the field site and eddy covariance instrumentation. (a) Layout of the field site. Dashed lines indicate the borders of the field treatments used in the larger 4-plot flux gradient (FG) study. Instrumentation for that study was housed in the indicated instrumentation trailer located in the center of the field. Solid lines outline the entire field. (b) Wind rose of the wind speeds and direction observed during the field study. (c) Configuration of the analyzers at the EC tower; A – TDLAS-LN, B – TDLAS-TE, C – EC155, D - reference gases for the $N_2O$ analyzers. (d) schematic of TDLAS-LN sampling system with inline filter and multi-tube dryer. (e) schematic of TDLAS-TE sampling system with vortex intake and single-tube dryer.

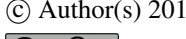



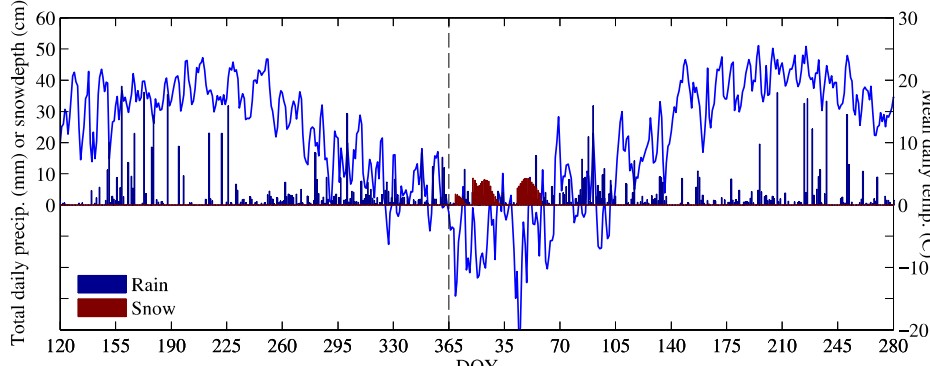

**Figure 2**: Daily precipitation and temperature for 2015 and 2016. Rain values (mm) are the total rainfall per day, snow values (cm) are the depth of snow present.

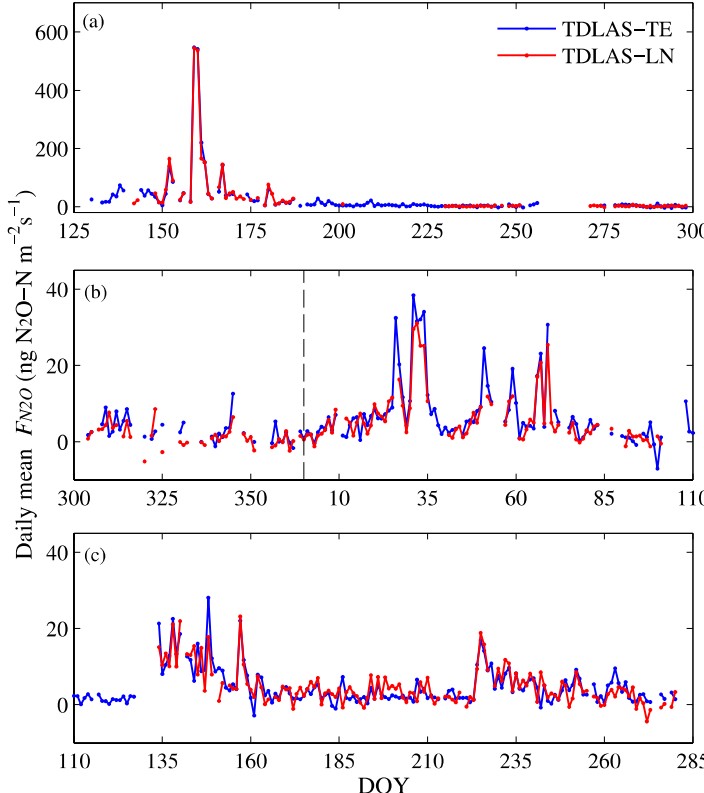

**Figure 3**: Mean daily fluxes of N$_2$O-N measured by each analyzer. Figure (a) shows the period of high N$_2$O fluxes in the post-fertilization period of 2015 and the fluxes measured throughout the growing season. Figure (b) shows the fluxes measured during the post-harvest period of 2015 and winter of 2016. The dotted line in (b) demarcates the separation between years 2015 and 2016. Figure (c) shows fluxes measured from early spring to the end of the growing season of 2016.





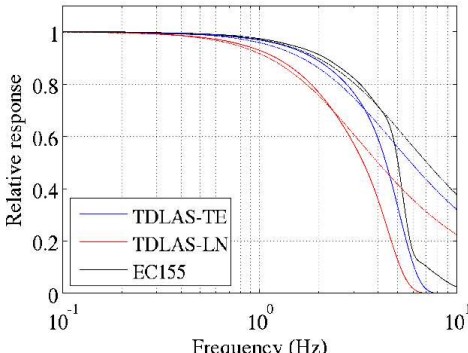

**Figure 4**: Results of the frequency response test of TDLAS-LN and TDLAS-TE for $N_2O$ and the EC155 ($CO_2$) under normal operating conditions (solid lines), dashed lines are modeled $T_{FR}$ calculated using the measured $f_c$ in equation 1.

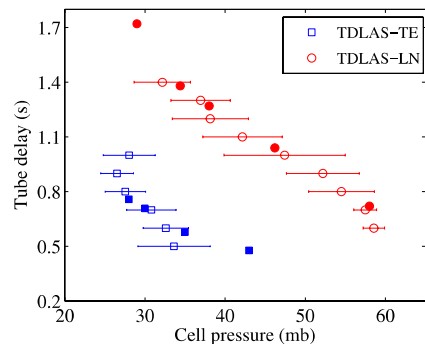

**Figure 5**: Tube delay ($t_{del}$) determined from the frequency response test (solid symbols) and from the cross-correlation between the $CO_2$ and $w$ (open symbols) across the range of operating pressures. Error bars are the standard deviation of the mean cell pressure associated with each value.





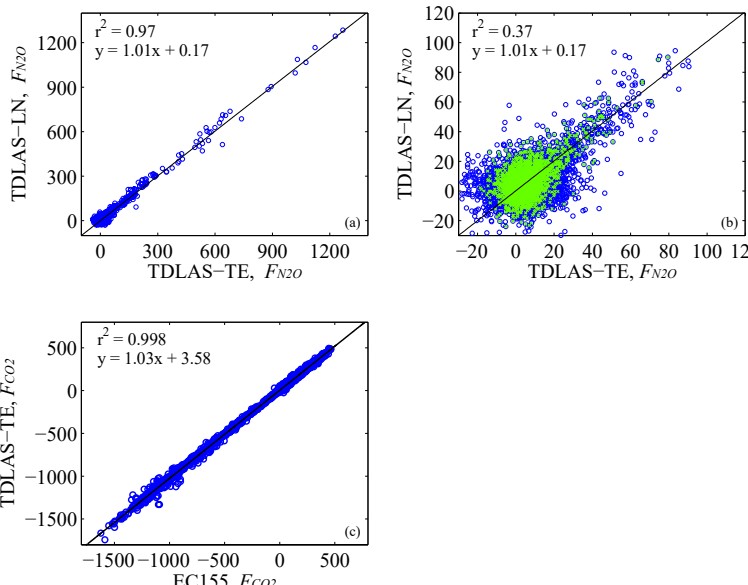

**Figure 6**: Comparison of 30 min fluxes measured by each analyzer. Figure (a) displays the comparison of $N_2O$-N fluxes (ng $N_2O$-N $m^{-2}s^{-1}$) for the entire dataset, (b) shows the same $N_2O$-N flux dataset (ng $N_2O$-N $m^{-2}s^{-1}$) but centered on the low flux values. The green 'x' markers in Figure (b) are the fluxes measured when the standard deviation of background $N_2O$ concentrations for both analyzers was less than 2 ppb $N_2O$. Figure (c) is the comparison of $F_{CO2}$ between TDLAS-TE and EC155 (μg $CO_2$ $m^{-2}s^{-1}$).

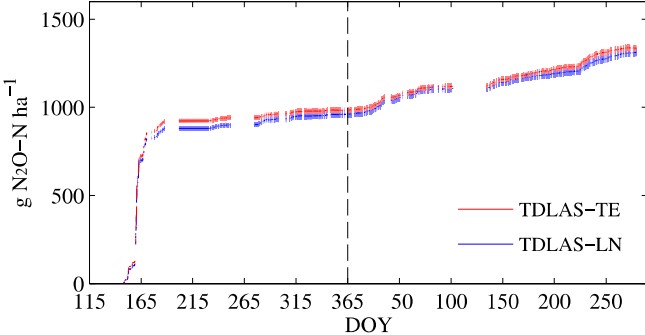

**Figure 7**: Cumulative $N_2O$-N emissions. Shaded areas represent the $p = 0.05$ confidence intervals calculated using the random error calculations.