# Peer review of "Evaluation of a lower-powered analyser and sampling system for eddy-covariance measurements of nitrous oxide fluxes"

_Atmospheric Measurement Techniques, 2017_

## Referee Comment (RC1) · Anonymous Referee #1 · 21 Aug 2017

General Comments

This manuscript presents an extensive study of the performance of a modified version of a tunable diode laser system for measuring N2O fluxes using eddy covariance. The paper is very well written with an excellent description of the Methods and Results. It is easy to follow and will be of general interest to the flux measurement community. The Discussion is appropriate, covers the relevant literature, and reflects the current state of knowledge of our science.

This paper should be published. My specific comments below are quite minor and can be easily accommodated by the authors.

[Figure]

Specific Comments

Lines 9 and 17. Avoid acronyms in the abstract such as EC and TDLAS.

Line 31. High r-square values do not tell us about the fluxes being similar; i.e. the slope or means could be quite different. Can you give an additional statistic to tell us how close the comparison was (e.g., regression slope or % difference between the mean values)?

Line 46. The word "significant" is not quite right here; perhaps you mean a sufficient spatial sample?

Line 48. So if we remove the acronyms in the Abstract, define "EC" here.

Line 90. Some references here, such as Wagner-Riddle 2017 reported on flux-gradient measurements, yet this discussion is about EC.

Line 306. Velocity instead of speed for "w".

Line 374. Perhaps "freezing" instead of "zero" so that the temperature units don't matter?

Line 459. Were the coefficients the same for the TE and LN comparisons, or is this a typo? If they were identical, perhaps reword.

Line 474. This paragraph is about EC measurements of N2O but the 2 references are more about CO2. Are they appropriate?

Figure 4. Units should be given for frequency on x-axis.

---

## Referee Comment (RC2) · M. Yang (Referee) · 10 Oct 2017

In this paper, the authors described a new N2O flux sensor that has a faster response and uses less power than previous sensors. This new sensors was deployed side-by-side against previous sensors and the fluxes of N2O and CO2 from an agricultural field over a growing season were compared, yielding very favourable results. This is a very well written paper and a useful reference for anyone working with closed-path eddy covariance sensors, especially from the perspective of choosing the optimal flow/cell volume, etc. I think the paper could be published as is but it can be improved further if the authors are able to incorporate my suggestions.

[Figure]

1. Being a method paper, it'd be useful to see some cospectra (in order to be convinced that all of the fluxes are captured). I recognize that small fluxes lead to noisy cospectra, but the authors could look at the brief period(s) when the N2O fluxes were very large. Do all the sensors show agreement?

This would also enable a direct comparison of high frequency flux loss between the frequency response method (which the authors have focused on so far) and the cospectral/ogive method.

2. The authors have reported frequency response times, but what kind of flux loss do they amount to under typical atmospheric conditions? The authors have presented the equations to estimate this but haven't provided any numbers (that I could see). This information would be informative for readers.

3. It would be useful to know what the flux detection limits are. There is a lot of interest now in knowing the emission of N2O from the ocean (especially low-oxygen regions). However per m2 the N2O flux over the ocean is typically orders of magnitude lower than fluxes over land. Would this sensor be able to detect oceanic N2O flux?

There are multiple ways of estimating the flux detection limits. See this paper as an example: https://www.atmos-meas-tech.net/9/5509/2016/

4. That the sensor is sensitive to motion (generated by wind) is a bit concerning. Any idea why and any effort in trying to reduce this sensitivity (rather than just securing the sensor better/in a more sheltered location)?
* * *

---

## Author Comment (AC1) · 2 Dec 2017

Technical corrections from Reviewer 1 were addressed as follows and were incorporated in the current version of the discussion paper:

Lines 9 and 17. Avoid acronyms in the abstract such as EC and TDLAS.

*Removed acronyms EC and TDLAS from the abstract, replaced with eddy-covariance or tunable diode laser absorption spectrometer.*

Line 31. High r-square values do not tell us about the fluxes being similar; i.e. the slope or means could be quite different. Can you give an additional statistic to tell us

how close the comparison was (e.g., regression slope or

*Included the regression slopes (p1, l24; p1 l26)*

Line 46. The word "significant" is not quite right here; perhaps you mean a sufficient spatial sample?

*Removed "significant", changed sentence to "This sporadic nature of $F_{N2O}$ necessitates continuous measurements covering areas large enough to capture the spatial heterogeneity of the fluxes to sufficiently quantify total emissions from agricultural systems"*

Line 48. So if we remove the acronyms in the Abstract, define "EC" here.

*Defined EC at line 9 page 2.*

Line 90. Some references here, such as Wagner-Riddle 2017 reported on flux-gradient measurements, yet this discussion is about EC.

*Removed the reference (p2, l13)*

Line 306. Velocity instead of speed for "w".

*Changed speed to velocity (p8, l25)*

Line 374. Perhaps "freezing" instead of "zero" so that the temperature units don't matter?

*Changed zero to freezing (p10, l22)*

Line 459. Were the coefficients the same for the TE and LN comparisons, or is this a typo? If they were identical, perhaps reword.

*Clarified that the comparisons were between TLDAS-TE and EC-155, and TDLAS-LN and EC-155 (p12, l24-26)*

Line 474. This paragraph is about EC measurements of $N_2O$ but the 2 references are

more about $CO_2$. Are they appropriate?

*Clarified that the data coverage refers to the data captured after standard EC-filtering for wind speeds and direction, which applies to both $CO_2$ and $H_2O$ EC fluxes (p13 l6)*

Figure 4. Units should be given for frequency on x-axis.

*Added units.*

---

## Author Comment (AC2) · 2 Dec 2017

1. Being a method paper, it'd be useful to see some cospectra (in order to be convinced that all of the fluxes are captured). I recognize that small fluxes lead to noisy cospectra, but the authors could look at the brief period(s) when the $N_2O$ fluxes were very large. Do all the sensors show agreement? This would also enable a direct comparison of high frequency flux loss between the frequency response method (which the authors have focused on so far) and the cospectral/ogive method.

*A supplement (S2) was added to the manuscript to provide spectral figures for the time period when there was an $N_2O$ flux event. This showed that both TDLAS analyzers had*

[Figure]

*similar spectral characteristics, and that both analyzers when using a non-optimized intake system capture the majority of the flux information. The cutoff frequencies calculated from the cospectral analysis were greater than 2 Hz.*

*It is difficult to directly compare the cospectral and frequency-response methods of calculating the absolute values of $f_c$. The cospectral method is highly dependent on the adherence of the spectra to similarity scaling. It is also affected by sensor separation and by imperfect synchronization of the scalar with vertical wind. For these reasons we believe the impulse response test provides a more accurate estimate of $f_c$, and we have included cospectral analysis in a supplement rather than the main body of the paper.*

*Included in Results (line 11 page 17):*

*Response times calculated using measured cospectra are included in the Supplementary Materials.*

2. The authors have reported frequency response times, but what kind of flux loss do they amount to under typical atmospheric conditions? The authors have presented the equations to estimate this but haven't provided any numbers (that I could see). This information would be informative for readers.

*Included at line 3 of page 17 :*

*Spectral losses (equation 7) calculated with the $f_c$ were on average 5% of the measured flux for the TLDAS-TE and 7% for the TDLAS-LN.*

3. It would be useful to know what the flux detection limits are. There is a lot of interest now in knowing the emission of $N_2O$ from the ocean (especially low-oxygen regions). However per $m^2$ the $N_2O$ flux over the ocean is typically orders of magnitude lower than fluxes over land. Would this sensor be able to detect oceanic $N_2O$ flux? There are multiple ways of estimating the flux detection limits. See this paper as an example: https://www.atmos-meas-tech.net/9/5509/2016/

*Included in Methods (line 11 page 14):*

*Flux detection limits ($\delta F$) were calculated using the method described in Blomquist et al. (2010) and Yang et al. (2016) where the detection limit is calculated using the instrument noise and the variability of the concentration signal:*

$$\delta F = \frac{2\sigma_w}{\sqrt{T_f}} \left[ \sigma_{C_a}^2 \tau_{wc} + \frac{\varphi_{C_n}}{4} \right]^{1/2} [8]$$

*such that $\sigma_w$ is the standard deviation of $w$, $T_f$ is the flux-averaging period, $\sigma_{C_a}^2$ is the ambient variance of the concentration signal, $\tau_{wc}$ is the integral timescale for the ambient concentration variance, and $\varphi_{C_n}$ is the band-limited analyzer noise. The $\varphi_{C_n}$ was calculated as the mean of the variance spectra from 1 to 5 Hz. Values of $\sigma_{C_a}^2$ were calculated as the second point of the autocovariance of the concentration signal (Yang et al. 2016). Blomquist et al. (2010) estimated $\tau_{wc}$ using the peak frequency of the variance cospectrum. Peak frequencies here were modelled via the equation given in Horst (1997).*

*Included in Results section (line 8 page 18):*

*Figure 7 shows the mean variance spectra of the $N_2O$ signal from each TDLAS analyzer for a period of 10 days where no $N_2O$ fluxes were observed (29 June 2016 to 9 July 2016). The variance spectra of both analyzers were dominated by instrument noise as $N_2O$ emissions were at background levels. Values of $\varphi_{C_n}$ were 0.28 ppb$^2$ Hz$^{-1}$ for the TDLAS-TE and 0.22 ppb$^2$ Hz$^{-1}$ for the TDLAS-LN. When considering only instrument noise, detection limits with mean conditions of U = 3.5 m s$^{-1}$, z = 4 m, and $\sigma_w$ = 0.5 m s$^{-1}$ were 6.6 ng $N_2O$-N m$^{-2}$s$^{-1}$ for the TDLAS-TE and 5.2 ng $N_2O$-N m$^{-2}$s$^{-1}$ for the TDLAS-LN. Incorporating signal noise increased $\delta F_{N2O}$ to 9.9 ng $N_2O$-N m$^{-2}$s$^{-1}$ for the TDLAS-TE and 19.6 ng $N_2O$-N m$^{-2}$s$^{-1}$ for the TDLAS-N. Despite the lower instrument noise, concentration signals of the TLDAS-LN were less steady than the TLDAS-TE during the period evaluated ($\sigma_{C_a}^2$ of 0.27 ppb$^2$ and 0.042 ppb$^2$, respectively).*

Figure 7: Mean $N_2O$ variance spectra of each TDLAS analyzer from 29 June 2016 to 9 July 2016.

4. That the sensor is sensitive to motion (generated by wind) is a bit concerning. Any idea why and any effort in trying to reduce this sensitivity (rather than just securing the sensor better/in a more sheltered location)?

*This motion sensitivity is caused by Fabry-Perot interference, a consequence of unwanted reflections within the optical path. This optical interference modulates the background of the spectral scan, causing an offset error in the concentration measurement. The optical system is not perfectly rigid, which allows wind-driven vibrations to introduce very small changes in optical alignment. These changes in alignment shift the position and shape of the optical interference pattern, thereby changing the offset error over time. However, a simple baffle along the side of the analyzer facing the mean wind direction, such in the way that the TDLAS-LN sheltered the TDLAS-TE, is sufficient shelter from strong winds. Compared to other EC-capable $N_2O$ analyzers that require larger weather-proofed, temperature controlled enclosures, a baffle is a minor addition to an EC tower.*

*Included in Discussion (line 8 page 19):*

*A simple baffle the height of the analyzer (0.55 m) along the side of the analyzer facing the mean wind direction would provide sufficient shelter from strong winds (U > 5 m $s^{-1}$).*

Please also note the supplement to this comment:
https://www.atmos-meas-tech-discuss.net/amt-2017-169/amt-2017-169-AC2-supplement.pdf

[Figure]

Figure 7: Mean $N_2O$ variance spectra of each TDLAS analyzer from 29 June 2016 to 9 July 2016.

**Fig. 1.**

**Supplement:**

S2. Spectral Analysis

Cutoff frequencies of EC systems can be evaluated using the ratio of the spectra of a scalar of interest to that of an assumed unattenuated temperature spectra ($f_{c,s}$). This involves computing ensemble-averaged spectra or cospectra using several hours of data with conditions that warrant scalar similarity (i.e., stationary, sunny conditions with positive sensible heat fluxes, moderate winds, and a high scalar flux (Aubinet et al., 2012)). Many studies using this empirical method select data from one ideal day meeting these conditions (Detto et al., 2011; Peltola et al., 2014; Rannik et al., 2015) to calculate $f_{c,s}$ as a metric for characterizing EC systems. Unfortunately, the low $N_2O$ fluxes observed in this study prevented an accurate analysis of the frequency response of the optimized TDLAS-TE system via spectral ratios.

One day of the study period had adequate conditions for spectral analysis (8 June 2015), i.e., consecutive 30 min periods with stationary signals, moderate winds (2 m·s$^{-1}$ < $U$ < 4 m·s$^{-1}$), strong fluxes, $|F_{N2O}|$ > 400 ng m$^{-2}$·s$^{-1}$), and positive sensible heat fluxes ($H$ > 25 Wm$^{-2}$). At that time both $N_2O$ analyzers were operating using the older-style sampling systems with nominal flows and pressures (58 and 59 mb for TDLAS-LN and TDLAS-TE, respectively). Ensemble variance spectra and cospectra of the $N_2O$ signals were calculated using data from 13:00 to 17:00 on 8 June 2015. Cospectral analysis for the EC155 used data from a period with strong $CO_2$ signals ($|F_{CO2}|$ > 500 µg m$^2$s$^{-1}$, 12:00 to 16:00 on 19 Aug 2015).

Spectral cutoff frequencies were calculated by fitting the cospectral ratios of the concentration signals and sonic temperature ($T_s$) to a first-order transfer function ($T$):

$$T = \frac{1}{\sqrt{1+\left(\frac{f}{f_{c,s}}\right)^2}}$$  [S1]

Results of the spectral analyses are displayed in Fig. S1. Spectra from each $N_2O$ analyzer (Fig. S1a) were similar up to ~1 Hz where the TDLAS-TE noise became observable. Fig. S1b shows very good agreement between the $N_2O$ cospectra as measured by each analyzer. Cospectral ratios

of both analyzers were similar (Fig. S1c) and demonstrated that both analyzers captured the majority of the $N_2O$ flux when using the older-style sampling system. $CO_2$ spectral analysis results for the EC155 are shown in Figs. S1 d, e, and f. Cutoff frequencies calculated by fitting equation S2.1 to the ratio of $Co(f)_{wN2O}$ to $Co(f)_{wT}$ gave $f_{c,s}$ = 2.5 Hz for the TDLAS-TE, $f_{c,s}$ = 2.7 Hz for the TDLAS-LN, and 3.05 Hz for the EC155. These results are not directly comparable to the in situ frequency response test because the cospectral analysis includes other losses, such as sensor separation and synchronization.

[Figure]

Figure S1. Results of the spectral analysis. Normalized variance spectra (Fig. S1a) and normalized cospectra (Fig. S1b), and cospectral ratios of $N_2O$ (Fig. S1c) measured by the TDLAS-TE and TDLAS-LN for the period with strong $F_{N2O}$ (13:00 to 17:00 local time on 8 June 2015, $zL^{-1}$ = -0.045, $u$ = 3.55 ms$^{-1}$). Figures d-f: Normalized variance spectra (Fig. S1c), normalized cospectra (Fig. S1d), and cospectral ratios of $CO_2$ (Fig. S1c) measured by the EC155 during a period of strong $CO_2$ flux (12:00 to 16:00 local time on 19 Aug 2015).

References

Aubinet, M., Vesala, T., and Papale, D. (Eds.): Eddy Covariance: A Practical Guide to Measurement and Data Analysis. Springer, 2012.

Detto, M., Verfaillie, J., Anderson, F., Xu, L., and Baldocchi, D.: Comparing laser-based open- and closed-path gas analyzers to measure methane fluxes using the eddy covariance method, Agricultural and Forest Meteorology, 151, 10, 1312-1324, 2011.

Peltola, O., Hensen, A., Helfter, C., Belelli Marchesini, L., Bosveld, F. C., van den Bulk, W. C. M., Elbers, J. A., Haapanala, S., Holst, J., Laurila, T., Lindroth, A., Nemitz, E., Röckmann, T., Vermeulen, A. T., and Mammarella, I.: Evaluating the performance of commonly used gas analysers for methane eddy covariance flux measurements: the InGOS inter-comparison field experiment, Biogeosciences, 11, 12, 3163-3186, 2014.

Rannik, U., Haapanala, S., Shurpali, N. J., Mammarella, I., Lind, S., Hyvönen, N., Peltola, O., Zahniser, M., Martikainen, P. J., and Vesala, T.: Intercomparison of fast response commercial gas analysers for nitrous oxide flux measurements under field conditions, Biogeosciences, 12, 2, 415-432, 2015.

---

## Author Response (AR1)

**Response to Reviewer #1:**

Technical corrections from Reviewer 1 were addressed as follows and are incorporated in the current version of the discussion paper. Comments are included in the marked-up manuscript below to identify the locations where changes were made.

**Lines 9 and 17. Avoid acronyms in the abstract such as EC and TDLAS.**

*Removed acronyms EC and TDLAS from the abstract, replaced with eddy-covariance or tunable diode laser absorption spectrometer.*

**Line 31. High r-square values do not tell us about the fluxes being similar; i.e. the slope or means could be quite different. Can you give an additional statistic to tell us how close the comparison was (e.g., regression slope or % difference between the mean values)?**

*Included the regression slopes (page 2, line 1; page 2 line 3)*

**Line 46. The word "significant" is not quite right here; perhaps you mean a sufficient spatial sample?**

*Removed "significant", changed sentence to "This sporadic nature of $F_{N2O}$ necessitates continuous measurements covering areas large enough to capture the spatial heterogeneity of the fluxes to sufficiently quantify total emissions from agricultural systems" (page 2 line 14)*

**Line 48. So if we remove the acronyms in the Abstract, define "EC" here.**

*Defined EC at page 2, line 17.*

**Line 90. Some references here, such as Wagner-Riddle 2017 reported on flux-gradient measurements, yet this discussion is about EC.**

*Removed the reference (page 2, line 20)*

**Line 306. Velocity instead of speed for "w".**

*Changed speed to velocity (page 12, line 2)*

**Line 374. Perhaps "freezing" instead of "zero" so that the temperature units don't matter?**

*Changed zero to freezing (page 15, line 2)*

**Line 459. Were the coefficients the same for the TE and LN comparisons, or is this a typo? If they were identical, perhaps reword.**

*Clarified that the comparisons were between TLDAS-TE and EC-155, and TDLAS-LN and EC-155 (page 18, line 3)*

**Line 474. This paragraph is about EC measurements of N2O but the 2 references are more about CO2. Are they appropriate?**

*Clarified that the data coverage refers to the data captured after standard EC-filtering for wind speeds and direction, which applies to both $CO_2$ and $H_2O$ EC fluxes (page 18 line 3)*

**Figure 4. Units should be given for frequency on x-axis.**

*Added units.*

**Response to Reviewer 2:**

**1. Being a method paper, it'd be useful to see some cospectra (in order to be convinced that all of the fluxes are captured). I recognize that small fluxes lead to noisy cospectra, but the authors could look at the brief period(s) when the $N_2O$ fluxes were very large. Do all the sensors show agreement? This would also enable a direct comparison of high frequency flux loss between the frequency response method (which the authors have focused on so far) and the cospectral/ogive method.**

A supplement (S2) was added to the manuscript to provide spectral figures for the time period when there was an $N_2O$ flux event. This showed that both TDLAS analyzers had similar spectral characteristics, and that both analyzers when using a non-optimized intake system capture the majority of the flux information. The cutoff frequencies calculated from the cospectral analysis were greater than 2 Hz.

*Included in Discussion (page 20, line 6):*

However, it is difficult to directly compare the cospectral and frequency-response methods of calculating the absolute values of $f_c$. The cospectral method is highly dependent on the adherence of the spectra to similarity scaling. It is also affected by sensor separation and by imperfect synchronization of the scalar with vertical wind. For these reasons we believe the impulse response test provides a more accurate estimate of $f_c$, and we have included cospectral analysis in a supplement rather than the main body of the paper.

*Included in Results (page 17, line 11):*
Response times calculated using measured cospectra are included in the Supplementary Materials.

**2. The authors have reported frequency response times, but what kind of flux loss do they amount to under typical atmospheric conditions? The authors have presented the equations to estimate this but haven't provided any numbers (that I could see). This information would be informative for readers.**

*Included page 17, line 3:*
Spectral losses (equation 7) calculated with the $f_c$ were on average 5% of the measured flux for the TLDAS-TE and 7% for the TDLAS-LN.

**3. It would be useful to know what the flux detection limits are. There is a lot of interest now in knowing the emission of N2O from the ocean (especially low-oxygen regions).**
**However per $m^2$ the $N_2O$ flux over the ocean is typically orders of magnitude lower than fluxes over land. Would this sensor be able to detect oceanic $N_2O$ flux? There are multiple ways of**

**estimating the flux detection limits. See this paper as an example: https://www.atmos-meas-tech.net/9/5509/2016/**

*Included in Methods (page 14, line 12):*

Flux detection limits ($\delta F$) were calculated using the method described in Blomquist et al. (2010) and Yang et al. (2016) where the detection limit is calculated using the instrument noise and the variability of the concentration signal:

$$\delta F \ = \ \frac{2\sigma_w}{\sqrt{T_f}}\left[\sigma_{C_a}^2\ \tau_{wc} + \frac{\varphi C_n}{4}\right]^{1/2} \tag{8}$$

such that $\sigma_w$ is the standard deviation of $w$, $T_f$ is the flux-averaging period, $\sigma_{C_a}^2$ is the ambient variance of the concentration signal, $\tau_{wc}$ is the integral timescale for the ambient concentration variance, and $\varphi_{Cn}$ is the band-limited analyzer noise. The $\varphi_{Cn}$ was calculated as the mean of the variance spectra from 1 to 5 Hz. Values of $\sigma_{C_a}^2$ were calculated as the second point of the autocovariance of the concentration signal (Yang et al. 2016). Blomquist et al. (2010) estimated $\tau_{wc}$ using the peak frequency of the variance cospectrum. Peak frequencies here were modelled via the equation given in Horst (1997).

*Included in Results section (line 8 page 18):*

Figure 7 shows the mean variance spectra of the $N_2O$ signal from each TDLAS analyzer for a period of 10 days where no $N_2O$ fluxes were observed (29 June 2016 to 9 July 2016). The variance spectra of both analyzers were dominated by instrument noise as $N_2O$ emissions were at background levels. Values of $\varphi_{Cn}$ were 0.28 $ppb^2\ Hz^{-1}$ for the TDLAS-TE and 0.22 $ppb^2\ Hz^{-1}$ for the TDLAS-LN. When considering only instrument noise, detection limits with mean conditions of $U = 3.5$ $ms^{-1}$, $z = 4$ m, and $\sigma_w = 0.5$ $ms^{-1}$ were 6.6 ng $N_2O$-N $m^{-2}s^{-1}$ for the TDLAS-TE and 5.2 ng $N_2O$-N $m^{-2}s^{-1}$ for the TDLAS-LN. Incorporating signal noise increased $\delta F_{N2O}$ to 9.93 ng $N_2O$-N $m^{-2}s^{-1}$ for the TDLAS-TE and 19.6 ng $N_2O$-N $m^{-2}s^{-1}$ for the TDLAS-N. Despite the lower instrument noise, concentration signals of the TLDAS-LN were less steady than the TLDAS-TE during the period evaluated ($\sigma_{C_a}^2$ of 0.27 $ppb^2$ and 0.042 $ppb^2$, respectively).

[Figure]

**Figure 7:** Mean N$_2$O variance spectra of each TDLAS analyzer from 29 June 2016 to 9 July 2016.

**4. That the sensor is sensitive to motion (generated by wind) is a bit concerning. Any idea why and any effort in trying to reduce this sensitivity (rather than just securing the sensor better/in a more sheltered location)?**

This motion sensitivity is caused by Fabry-Perot interference, a consequence of unwanted reflections within the optical path. This optical interference modulates the background of the spectral scan, causing an offset error in the concentration measurement. The optical system is not perfectly rigid, which allows wind-driven vibrations to introduce very small changes in optical alignment. These changes in alignment shift the position and shape of the optical interference pattern, thereby changing the offset error over time. However, a simple baffle along the side of the analyzer facing the mean wind direction, such in the way that the TDLAS-LN sheltered the TDLAS-TE, is sufficient shelter from strong winds. Compared to other EC-capable N$_2$O analyzers that require larger weather-proofed, temperature controlled enclosures, a baffle is a minor addition to an EC tower.

[revised manuscript text omitted]

---

## Author Response (AR2)

**Response to Comments from the Editor:**

Having a system for N2O field scale flux measurements with better frequency response than previous systems and at the same time lower sample pump power consumption is a step forward with regard to remote agricultural areas where high-quality power supply is not always available. The technical characterization of the system is rigorous and well done. Including the suggestions of Referee #2 regarding spectral losses and flux detection limit has further improved the paper.

The study is a valuable contribution to the eddy flux community and can be published subject to only a few minor additions:

- On page 18, lines 20-24 and in Fig.8 it should again be indicated that no gap filling was used. Otherwise these total emission numbers do not make sense and may mislead the reader when trying to compare these figures with other studies.

*Added to Results (page 18 line 17):*
*No gap-filling was applied for the comparison of cumulated sums between analyzers.*

*Changed Fig. 8 caption to:*
*Cumulative (not gap-filled) $N_2O$-N emissions.*

- Accurate quantification of N losses from fertilized agricultural land is crucial when trying to optimize fertilizer type, amount, application technique, and yield while at the same time keeping N emissions on a low level. I'm missing a rough estimation of the ratio of N emitted to N added (i.e. emission factor) in order to further assess the field applicability of the system. Is information about amount of fertilizer added (N content) available? For this purpose, a simple gap filling routine should be performed (something like mean diurnal variation). Do the results make sense and are they comparable to other studies (maybe even to chamber measurements at the same site)? If available, these information should be added on page 15, lines 10-25.

*Added to Methods (page 5 line 17):*
*at a rate of 150 kg ha$^{-1}$*

*Added to Methods (page 12 line 14):*
*Cumulative gap-filled seasonal emission rates (planting to harvest) were calculated for each year. Daily mean emissions were estimated by extrapolating the mean of the available 30 min fluxes for each day to g $N_2O$-N ha$^{-1}$. Linear interpolation was used to fill periods with missing data (Abalos et al., 2015).*

*Added to Results (page 15 line 19):*
*Mean gap-filled daily emissions as measured by each analyzer were averaged together. This gave total seasonal (planting to harvest) cumulative emissions of 3.34 kg $N_2O$-N ha$^{-1}$ and 1.20 kg $N_2O$-N ha$^{-1}$ for the growing seasons of 2015 and 2016, respectively. This is*

*within the range of seasonal emissions typically observed at this site (Wagner-Riddle et al., 2007; Abalos et al., 2015) and represented a loss of 2.2% and 0.8% of the applied N.*

- Section 3.4 should be 3.3.

*Changed.*